# Safe Robust Adaptive Motion Control for Underactuated Marine Robots

**DOI:** 10.3390/s24123974

**Published:** 2024-06-19

**Authors:** G. Reza Nazmara, A. Pedro Aguiar

**Affiliations:** SYSTEC, ARISE, and Department of Electrical and Computer Engineering, Faculty of Engineering, University of Porto, 4200-465 Porto, Portugal; pedro.aguiar@fe.up.pt

**Keywords:** marine robots, funnel control, backstepping control, fuzzy systems, finite-time stability

## Abstract

This article presents an innovative approach to the design of a safe adaptive backstepping control system. Tailored specifically for underactuated marine robots, the system utilizes simple sensors for enhanced practicality and efficiency. Given their operation in diverse oceanic environments fraught with various sources of uncertainties, ensuring the system’s safe and robust behavior holds paramount importance in the control literature. To address this concern, this paper introduces a control strategy designed to ensure robustness at both the kinematic and dynamic levels. By emphasizing the compensation for the system uncertainties, the design integrates a straightforward fuzzy system structure. To further ensure the system’s safety, a funnel surface is defined, followed by the design of a suitable nonlinear sliding surface as a function of the funnel and tracking error. Using Lyapunov theory, the study formally establishes the Semi-globally Practically Finite-time Stability of the closed-loop system, validated through simulations conducted on underactuated marine robots.

## 1. Introduction

Autonomous and self-mission robots, particularly those designated for unmanned operations in the ocean such as marine vehicles, accompanied by suitable control schemes, have long stood as compelling research subjects, captivating the attention of numerous researchers. The resulting advancements have led to significant progress, providing valuable insights into the field. These insights could find numerous applications, such as marine research, subsea pipeline monitoring, lifesaving operations, oil and gas exploration, and many other missions [1,2,3]. However, despite the considerable strides made, the complexities surrounding underwater robotics persist. Challenges such as external disturbances, the unpredictable nature of fluid dynamics, navigation, energy management, environmental hazards and obstacles, underwater communication, autonomous decision-making and adaptation, as well as data processing, remain formidable obstacles. While considerable research has been dedicated to addressing the aforementioned issues, numerous challenges persist in the shadows. Therefore, additional research and discourse are imperative to develop robust and reliable algorithms capable of meeting the requisite standards and illuminating the intricacies of these challenges. In addition, autonomous marine vehicles encounter challenges categorized as underactuated, wherein the number of available actuators is fewer than the degrees of freedom. This circumstance adds further complexity to the design of control systems [3,4].

To meet these goals, schemes are needed to be implemented that are stable and safe. There are different ways to ensure the stability and safety of the system. One strong way to ensure formal stability is to use the Control Lyapunov functions [1,2,3]. These functions can provide a rigorous mathematical foundation for stability analysis. But this technique cannot alone ensure safety. To address safety, Control Barrier Functions (CBF) [5,6] and Funnel Control (FC) [7,8] have been proposed. One difficulty revolving around CBF is finding a suitable barrier function. Another issue regarding CBF is that it involves solving optimization problems as part of the control design process [5,6], which may be time-consuming and could be problematic in those applications that a fast and online reaction concerns.

Conversely, in FC [7,8], the defined boundary variable can easily mixed with nonlinear sliding surfaces and be used in Lyapunov functions in an outlier supervisory-like loop for monitoring some signals or confining some internal signals to achieve the requirement of safety and stability, without having to solve the optimization problems.

To address different uncertainties a strong approximation may be used. Among different approaches, Fuzzy Systems due to their special mathematical structure can be flexible enough to attract great attention to providing a strategy that could generally work for nonlinear systems and handling complex systems while being computationally efficient [2,8,9,10]. To overcome unmodelled dynamics, parametric uncertainties, and external disturbances, Sliding Mode Control (SMC) [1,11] proved to be effective enough to address these difficulties. However, from the practical point of view, the convergence time always matters, and thereby with ordinary SMC, designers may have less control over the rate of convergence. To achieve this objective, Terminal Sliding Mode Control (TSMC) has been developed, capitalizing on the robustness inherent in the original SMC-based methodologies, to ensure convergence within a finite time frame [3]. In this reference, a TSMC approach was devised for the trajectory tracking of underactuated underwater robots. Additionally, in [12], an innovative study introduced an adaptive finite-time impedance backstepping control scheme tailored for robot manipulators operating within unfamiliar environments. In practical terms, the inclusion of certain types of functions within the nonlinear sliding surface, such as the utilization of a sign function [1], can potentially lead to chattering in certain scenarios. To mitigate these challenges, the TSMC approach, as outlined in [3], has replaced the sign function with a saturation function, effectively resolving the potential issue of chattering. Another challenge associated with SMC pertains to the development of methodologies that do not impose restrictions on the rotational motion of the robot. For instance, this limitation is evident in the research cited in [13,14], as reported in [3].

Unlike many proposed control approaches like what was conducted on motion tracking [13,14] which have the limitation of being only valid for tracking special cases of reference trajectories, our proposed method can be used for a wide range of trajectories regardless of their shape. Moreover, compared to the safety methods developed in [5,6], in our case, we do not have to solve possible time-consuming optimizations since we introduce a novel boundary layer that utilizes a new soft funnel shape, contrasting with the approach in [7]. Additionally, unlike the methods discussed in [1,12], our proposed boundary layer does not incorporate a sign function within its structure, ensuring the robot’s signals remain within the safe set. In terms of feasibility and robustness, our proposed scheme surpasses much of the above research by considering a broader range of uncertainties through the utilization of a more straightforward fuzzy estimator compared to [2,8]. Moreover, the developed method does not use the direct feedback of the lateral velocity, eliminating the need for a costly lateral velocity sensor and thereby enhancing its economic viability. The primary contributions of this article, as an extended version of [15], are outlined as follows:Trajectory tracking of an underactuated marine vehicle is addressed amidst significant sources of uncertainties, encompassing time-varying external disturbances at both the Kinematics and Dynamics levels, unmodelled dynamics, time-varying water current speeds, lateral velocities, and estimator errors;Robust funnel control is proposed for trajectory tracking with formal stability guarantees, ensuring finite-time stability and safety;Control design eliminates the necessity for exact knowledge of the robot dynamics, enhancing design simplicity and feasibility through the utilization of straightforward feedback sensors. For instance, there is no requirement for lateral sensors or other complex sensors such as acceleration feedback;Leveraging developed mathematics and formulations, along with a TSMC and novel funnel boundary shape, designers can tailor to diverse requirements and uphold various safety measures across different initial conditions;Implementation of a fuzzy estimator facilitates the estimation of unmodelled dynamics in the robot, employing a simple design and straightforward inputs;Compensation for the imprecision of fuzzy systems is achieved through the integration of a robust controller.

The remainder of this article is organized as follows: Section 2 delves into the Kinematics and Dynamics of marine vehicles within a two-dimensional framework. In Section 3, the architecture of the proposed control technique is detailed, followed by a discussion on stability analysis in Section 4. Section 5 presents simulation results conducted on challenging scenarios, while Section 6 concludes with final remarks.

## 2. Robot Kinematics and Dynamics

In this section, we delve into the Kinematics and Dynamics of marine vehicles. Matrices and vectors are denoted in bold throughout this article for clarity.

By employing a two-dimensional representation of the robot’s movement path on a plane, the Kinematics of the marine vehicle can be expressed through the following equation (see, e.g., [16,17]): (1)$$\dot{x}=\left[\begin{array}{c} \dot{x}\\ \dot{y}\\ \dot{\psi } \end{array}\right]=\left[\begin{array}{ccc} \cos(\psi ) & -\sin(\psi ) & 0\\ \sin(\psi ) & \cos(\psi ) & 0\\ 0 & 0 & 1 \end{array}\right]\left[\begin{array}{c} u_{r}+u_{c}\\ v_{r}+v_{c}\\ r \end{array}\right]+\left[\begin{array}{c} \tau_{d_{kx}}\\ \tau_{d_{ky}}\\ \tau_{d_{k\psi }} \end{array}\right]$$
where *x*, *y*, and ψ denote the displacement along the *x* and *y* axes and the heading angle of the vehicle, respectively. Other signals including $u_{r}$, $v_{r}$, and *r* represent the surge, sway, and rotational speeds of the vehicle, while $u_{c}$ and $v_{c}$ denote the water current along surge and sway, respectively. Let us consider the vector $\tau_{d_{K}}$ representing the external disturbances exerted in displacements along the x and y axes as $\tau_{d_{K}}={\left[\begin{array}{cc} \tau_{d_{kx}} & \tau_{d_{ky}} \end{array}\right]}^{⊤}$.

The dynamics equation of the robot, which encapsulates motion along the surge, sway, and rotational axes, can be formulated as follows [16,17]:
(2a)$$\begin{array}{c} m_{u}{\dot{u}}_{r}-m_{v}v_{r}r+d_{u_{r}}u_{r}-\tau_{d_{u_{r}}}=\tau_{u}, \end{array}$$
(2b)$$\begin{array}{c} m_{v}{\dot{v}}_{r}+m_{u}u_{r}r+d_{v_{r}}v_{r}-\tau_{d_{v_{r}}}=0, \end{array}$$
(2c)$$\begin{array}{c} m_{r}\dot{r}-\left(m_{u}-m_{v}\right)u_{r}v_{r}+d_{r}r-\tau_{d_{r}}=\tau_{r} \end{array}$$
Here, $m_{u}$, $m_{v}$, and $m_{r}$ are terms of the inertia matrix including hydrodynamic added mass information, while $d_{u_{r}}$, $d_{v_{r}}$, and $d_{r}$ characterize the hydrodynamic damping effects. Other signals, such as $\tau_{d_{u_{r}}}$, $\tau_{d_{v_{r}}}$, and $\tau_{d_{r}}$, represent external disturbances along the surge, lateral, and rotational axes, respectively. The signals $\tau_{u}$ and $\tau_{r}$ denote the control signals applied in the surge and rotational directions, respectively. By combining (Equation 1) and (2), and considering the position vector $p={\left[\begin{array}{cc} x & y \end{array}\right]}^{⊤}$, the following is derived:(3)$$DR^{⊤}\ddot{p}+\left(D{\dot{R}}^{⊤}+CR^{⊤}\right)\dot{p}+D\dot{\lambda }+C\lambda -\tau_{d_{D}}-\left(DR^{⊤}{\dot{\tau }}_{d_{K}}+\left(D{\dot{R}}^{⊤}+CR^{⊤}\right)\tau_{d_{K}}\right)=\tau$$
in which the matrices D, C, τ, and $\tau_{d_{D}}$ are defined as $D=\left[\begin{array}{cc} m_{u} & 0\\ 0 & m_{r} \end{array}\right]$,

$C=\left[\begin{array}{cc} d_{u_{r}} & -m_{v}v_{r}\\ -\left(m_{u}-m_{v}\right)v_{r} & d_{r} \end{array}\right]$, $\tau ={\left[\begin{array}{cc} \tau_{u_{r}} & \tau_{r} \end{array}\right]}^{⊤}$, and $\tau_{d_{D}}={\left[\begin{array}{cc} \tau_{d_{u_{r}}} & \tau_{d_{r}} \end{array}\right]}^{⊤}$, respectively. In (Equation 3), R is the rotation matrix defined as $R=\left[\begin{array}{cc} \cos(\psi ) & -\sin(\psi )\\ \sin(\psi ) & \cos(\psi ) \end{array}\right]$ and λ is formulated as: (4)$$\lambda =-{\left[\begin{array}{cc} u_{c} & -r+v_{r}+v_{c} \end{array}\right]}^{⊤}$$

Next, we introduce a new variable $q=R^{⊤}p$. In doing so, Equation (Equation 3) can be reformulated as follows: (5)$$\ddot{q}={\hat{D}}^{-1}\tau +\mu$$
in which μ is given by
(6)$$\begin{array}{c} \mu =-{\hat{D}}^{-1}\left(D\dot{\lambda }+C\lambda -\tau_{d_{D}}+\eta +\left(DR^{⊤}\ddot{R}+\left(D{\dot{R}}^{⊤}+CR^{⊤}\right)\dot{R}\right)q+\left(D-\hat{D}\right)\ddot{q}\right)\\ -{\hat{D}}^{-1}\left(2DR^{⊤}\dot{R}+D{\dot{R}}^{⊤}R+CR^{⊤}R\right)\dot{q} \end{array}$$
where η has the following components: (7)$$\eta =-DR^{⊤}{\dot{\tau }}_{d_{K}}-\left(D{\dot{R}}^{⊤}+CR^{⊤}\right)\tau_{d_{K}}$$

## 3. Proposed Controller

Before designing the Backstepping controller, it is imperative to establish a tracking error, defined as the disparity between the actual rotated path and its desired trajectory. Let us define the rotated tracking error as
(8)$$\xi =R^{⊤}\left(p-p_{d}\right),$$
where the vector $\xi ={\left[\begin{array}{cc} \xi_{1} & \xi_{2} \end{array}\right]}^{⊤}$ is composed of two elements. Equation (Equation 8) represents the rotated tracking error and contains information regarding both the linear displacement and rotational components.

In consideration of the controller’s safety measures, the following funnel variable is employed: (9)$$\chi_{i}=\xi_{i}^{2}{\left(\nu_{i}^{2}-\xi_{i}^{2}\right)}^{-1}$$
In this context, $\nu_{i}$ represents a function with a funnel shape that delineates a safe permitted area for the tracing error over time and in this paper is proposed as follows: (10)$$\nu_{i}=\left(\varrho_{0}-\varrho_{\infty }\right){\left(1+\beta t\right)}^{-1}+\varrho_{\infty }$$
Here, $\varrho_{0}$ and $\varrho_{\infty }$ represent the initial and steady values of $\nu_{i}$, respectively, while β is a positive constant chosen by the designer to control the funnel boundary. The following nonlinear sliding surface is proposed as follows: (11)$$S_{i}=\xi_{i}+\zeta_{i}\int \tanh\left(\chi_{i}^{2}\right)\xi_{i}d_{\tau }$$
where $\xi_{i}\neq 0$ and $\zeta_{i}>0$. In the proposed control method, the nonlinear sliding surface $x_{1i}=s_{i}$ and its time derivative $x_{2i}={\dot{s}}_{i}$ are considered the system variables. These variables are later demonstrated to serve as inputs for the fuzzy estimator. To calculate the time derivatives of $x_{1i}$ and $x_{2i}$, it would be straightforward if we first compute the time derivative of $\chi_{i}$. To this end, the time derivative of χ is given as the following equation: (12)$${\dot{\chi }}_{i}=\left(2{\nu_{i}}^{2}\xi_{i}{\dot{\xi }}_{i}-2\nu_{i}\xi_{i}^{2}{\dot{\nu }}_{i}\right){\left({\nu_{i}}^{2}-{\xi_{i}}^{2}\right)}^{-2}$$
Let us assign $A_{i}$ and $B_{i}$ as $A_{i}=\left(2{\nu_{i}}^{2}\xi_{i}\right){\left({\nu_{i}}^{2}-{\xi_{i}}^{2}\right)}^{-2}$ and $B_{i}=\left(2\nu_{i}\xi_{i}^{2}{\dot{\nu }}_{i}\right){\left({\nu_{i}}^{2}-{\xi_{i}}^{2}\right)}^{-2}$, then one can rewrite Equation (Equation 12) as follows: (13)$${\dot{\chi }}_{i}=A_{i}{\dot{\xi }}_{i}-B_{i}$$
Taking the time derivative of this equation yields the following equality: (14)$${\ddot{\chi }}_{i}=A_{i}{\ddot{\xi }}_{i}+{\dot{A}}_{i}{\dot{\xi }}_{i}-\dot{B_{i}}$$
The state-space representation can be expressed as follows:
(15a)$$\begin{array}{c} {\dot{x}}_{1i}=x_{2i}, \end{array}$$
(15b)$$\begin{array}{c} {\dot{x}}_{2i}={\ddot{\xi }}_{i}+2\zeta_{i}\dot{\chi }\chi_{i}{\mathrm{sech}}^{2}\left(\chi_{i}^{2}\right)\xi_{i}+\zeta_{i}\tanh\left(\chi_{i}^{2}\right){\dot{\xi }}_{i} \end{array}$$
By defining $\ddot{\xi }i$ as $\ddot{\xi }i={\ddot{q}}_{i}-{\ddot{q}}_{di}$ and utilizing Equation (Equation 5), Equation (15) can be rewritten as follows:
(16a)$$\begin{array}{c} {\dot{x}}_{1i}=x_{2i}, \end{array}$$
(16b)$$\begin{array}{c} {\dot{x}}_{2i}={\hat{D}}_{i}^{-1}\tau_{i}+{\bar{\mu }}_{i} \end{array}$$
in which ${\bar{\mu }}_{i}$ is given by
(17)$${\bar{\mu }}_{i}=\mu_{i}-{\ddot{q}}_{di}+2\zeta_{i}\dot{\chi }\chi_{i}{\mathrm{sech}}^{2}\left(\chi_{i}^{2}\right)\xi_{i}+\zeta_{i}\tanh\left(\chi_{i}^{2}\right){\dot{\xi }}_{i}$$

The state representation, as formulated in (16) with the uncertainty function (Equation 17), will be employed in designing the proposed control technique. This paper introduces the following control law: (18)$$\tau_{i}=-\hat{D}\left[x_{1i}+K_{1i}\left(2l-1\right)x_{2i}x_{1i}^{2l-2}+K_{2i}{\left(x_{2i}+k_{1i}x_{1i}^{2l-1}\right)}^{2l-1}\right]+\hat{D}\left[\tau_{ri}-{{\hat{\theta }}_{i}}^{⊤}\gamma_{i}\right]$$
Here, $K_{1i}$ and $K_{2i}$ are the control design parameters, which must be positive, *l* is a natural number, and $\tau_{ri}$ represents the robust term formulated as follows: (19)$$\tau_{ri}=-\left(x_{2i}+k_{1i}x_{1i}^{2l-1}\right){\hat{\rho }}^{2}{\left(|x_{2i}+k_{1i}x_{1i}^{2l-1}|\hat{\rho }+\delta_{ri}e^{-\sigma_{ri}t}\right)}^{-1}$$
Here, $\hat{\rho }$ represents the upper estimated bound of the modeling uncertainty using the fuzzy system. To prevent the denominator of Equation (Equation 19) from being zero, as evident in the formula, a small positive value $\delta_{ri}e^{-\sigma_{ri}t}$ is added to the denominator, and $\sigma_{ri}$ is another positive constant selected by designers. The adaptation procedure for the signals ${\hat{\theta }}_{i}$ and ${\hat{\rho }}_{i}$ in this paper is outlined by the following two equations: (20)$${\dot{\hat{\theta }}}_{i}=\beta_{i}\left(x_{2i}+k_{1i}x_{1i}^{2l-1}\right)\gamma_{i}-\sigma_{\theta i}{\hat{\theta }}_{i}$$
(21)$${\dot{\hat{\rho }}}_{i}=\alpha_{i}\left|x_{2i}+k_{1i}x_{1i}^{2l-1}\right|-\sigma_{\rho i}{\hat{\rho }}_{i}$$
where $\beta_{i}$ and $\alpha_{i}$ are the learning rates, and $\sigma_{\theta i}$ and $\sigma_{\rho i}$ are constant parameters chosen by designers. In Equation (Equation 18), ${{\hat{\theta }}_{i}}^{⊤}\gamma_{i}$ represents the output of the fuzzy system corresponding to the *i*-th estimator, which is intended to estimate the uncertainty. The fuzzy estimator structure in this article employs a product inference engine, singleton fuzzifier, center-average defuzzifier, and Gaussian membership functions [18]. Its architecture is as follows:(22)$${\hat{\bar{\mu }}}_{i}={{\hat{\theta }}_{i}}^{⊤}\gamma_{i}=\frac{\sum_{\iota_{i}=1}^{M_{i}}{{\bar{\theta }}_{i}}^{\iota_{i}}{\mu_{A_{1}}}^{\iota_{i}}\left(x_{1}\right){\mu_{A_{2}}}^{\iota_{i}}\left(x_{2}\right)}{\sum_{\iota_{i}=1}^{M_{i}}{\mu_{A_{1}}}^{\iota_{i}}\left(x_{1}\right){\mu_{A_{2}}}^{\iota_{i}}\left(x_{2}\right)}$$
In this context, $M_{i}$ denotes the number of fuzzy rules corresponding to each task-space variable, respectively, while ${\mu_{A_{j}}}^{\iota_{i}}\left(x_{j}\right)$ represents the membership function defined over the input range. In the realm of fuzzy systems, the degree of membership signifies the level to which an element is associated with a specific fuzzy set.

Considering the problem of two-dimensional motion control as outlined in the dynamic equation in this paper, it is necessary to employ two fuzzy estimators: one dedicated to compensating for uncertainties along the x-axis and the other along the y-axis. Each estimator is designed to accommodate two inputs. Based on these two fuzzy inputs, which are $S_{i}$ and ${\dot{S}}_{i}$, one can rewrite (Equation 22) as follows: (23)$${\hat{\bar{\mu }}}_{i}=\frac{\sum_{\iota_{i}=1}^{M_{i}}{{\bar{\theta }}_{i}}^{\iota_{i}}{\mu_{A_{1}}}^{\iota_{i}}\left(\xi_{i}+\zeta_{i}\int \tanh\left(\chi_{i}^{2}\right)\xi_{i}d_{\tau }\right){\mu_{A_{2}}}^{\iota_{i}}\left({\dot{\xi }}_{i}+\zeta_{i}\tanh\left(\chi_{i}^{2}\right)\xi_{i}\right)}{\sum_{\iota_{i}=1}^{M_{i}}{\mu_{A_{1}}}^{\iota_{i}}\left(\xi_{i}+\zeta_{i}\int \tanh\left(\chi_{i}^{2}\right)\xi_{i}d_{\tau }\right){\mu_{A_{2}}}^{\iota_{i}}\left({\dot{\xi }}_{i}+\zeta_{i}\tanh\left(\chi_{i}^{2}\right)\xi_{i}\right)}$$
Simplicity dictates that three membership functions, namely *N*, *Z*, and *P* as depicted in Figure 1, are assigned with the following mathematical equations for each fuzzy estimator’s input: (24)$$\mu_{N}\left(x_{i}\right)=\left\{\begin{array}{cc} 1 & \mathrm{if}x_{i}<-1,\\ e^{\frac{-{\left(x_{i}+1\right)}^{2}}{2\times 0.5^{2}}}\; & \mathrm{if}x_{i}\geq -1. \end{array}\right.$$
(25)$$\mu_{Z}\left(x_{i}\right)=e^{\frac{-x_{i}^{2}}{2\times 0.5^{2}}}$$
(26)$$\mu_{P}\left(x_{i}\right)=\left\{\begin{array}{cc} e^{\frac{-{\left(x_{i}-1\right)}^{2}}{2\times 0.5^{2}}} & \mathrm{if}x_{i}<1,\\ 1\; & \mathrm{if}x_{i}\geq 1. \end{array}\right.$$

To achieve a comprehensive understanding of the developed controller, the overall block diagram of the proposed method and the coordinate frames of the robot’s motion are illustrated in Figure 2, parts (a) and (b), respectively.

For designing the fuzzy system let us consider the following definitions for the membership functions and the set of rules:

**Definition** **1.**
*In the context of fuzzy systems, a membership function is considered normal if its maximum height value is one, as defined by [18], that is:*

(27)
$${\mu_{A_{i}}}^{\iota_{i}}\left(x_{i}\right)\leq 1$$

*Here, $x_{i}$ represents the i-th input for i=1 to $n_{i}$, where $n_{i}$ is the total number of inputs. $\iota_{i}$ denotes the rule index for the membership function associated with $x_{i}$, where $\iota_{i}=1$ to $M_{i}$, and $M_{i}$ is the total number of rules for the fuzzy set $A_{i}$. ${\mu_{A_{i}}}^{\iota_{i}}\left(x_{i}\right)$ stands for the membership function of input element $x_{i}$ in the fuzzy set $A_{i}$.*


**Definition** **2.**
*A collection of fuzzy IF-THEN rules is considered complete provided that for every $x_{i}\in U_{i}$, there is at least one rule in the fuzzy rule base, for instance, $\iota_{i}$, denoted as ${\mu_{A_{i}}}^{\iota_{i}}\left(x_{i}\right)$ in the format of (Equation 24)–(Equation 26), where:*

(28)
$${\mu_{A_{i}}}^{\iota_{i}}\left(x_{i}\right)\neq 0$$

*In other words, a membership function is considered complete if it covers the entire input range with appropriate overlap, ensuring that every point in the domain has at least a membership in one fuzzy set [18].*


According to Definition 1, Equations (Equation 24)–(Equation 26) are normal. Furthermore, designing based on Definition 2 implies that the denominator of Equation (Equation 23) is not zero.

## 4. Analyzing Control System Stability

### 4.1. Overall Stability of the System

This section aims to analyze the stability of the overall system, which consists of two subsystems based on the backstepping principles. To achieve this goal, the following information needs to be considered:

**Assumption** **1.**
*The external disturbances influencing the Kinematics, and Dynamics of the robots are considered to be bounded, that is,*

(29)
$$∥\left(\begin{array}{ccc} \tau_{d_{kx}}, & \tau_{d_{ky}}, & \tau_{d_{k\psi }} \end{array}\right)∥\leq \epsilon_{K}$$


(30)
$$∥(\begin{array}{ccc} \tau_{d_{u_{r}}}, & \tau_{d_{v_{r}}}, & \tau_{d_{r}} \end{array})∥\leq \epsilon_{D}$$

*where $\epsilon_{k}$ and $\epsilon_{D}$ are positive constants.*


**Assumption** **2.**
*The desired trajectory, $p_{d}$, is sufficiently smooth, and its time derivative is assumed to be readily available up to the required order [10].*


**Assumption** **3.**
*It is assumed that the water current velocity signals in the surge and sway directions are bounded.*

(31)
$$|u_{c}|\leq \epsilon_{u_{c}}$$


(32)
$$|v_{c}|\leq \epsilon_{v_{c}}$$



**Definition** **3**([7]). *To establish a particular stability criterion, we initially examine a nonlinear system subject to the following conditions*
(33)$$\dot{z}=N\left(z,u\right),\;N\left(0,0\right)=0,\;z\in R^{n},\;u\in R^{m}$$
*in which z signifies the state vector and u denotes the control input. The function N, which maps the domain from $R^{n+m}$ to $R^{n}$, is assumed to be continuous and defined within an open region around the origin. If there exists a neighborhood U encompassing the equilibrium point, and for each initial condition $z_{0}\in U$, there exists a positive constant ε along with a time-varying yet bounded parameter $\delta (\varepsilon ,z_{0})$ such that ∥z(t)∥<ε for all $t>t_{0}+\delta$, then the system described by (Equation 33) is termed to have Semi-Globally Practically Finite-Time Stability (SGPFS).*

**Proposition** **1.**
*Assuming that the vector z operates within the input universe denoted as U, a compact set within $R^{n}$, let us consider any real continuous function ${\bar{\mu }}_{i}\left(z\right)$ defined over this set and an arbitrary $\epsilon_{i}>0$. Under these conditions, it can be asserted that there is a fuzzy system, denoted as ${\hat{\bar{\mu }}}_{i}\left(z\right)$, structured with a product inference engine, singleton fuzzifier, center-average defuzzifier, and Gaussian membership functions. This system is designed such that the following relation holds:*

(34)
$$\underset{z\in A}{\sup}\left|{\hat{\bar{\mu }}}_{i}\left(z\right)-{\bar{\mu }}_{i}\left(z\right)\right|\leq \epsilon_{i}$$

*This inequality suggests that the fuzzy system mentioned serves as a universal approximation [18]. In this context, ${\hat{\bar{\mu }}}_{i}$ represents the output of the fuzzy system, and $\epsilon_{i}$ denotes both the precision of the fuzzy system and the upper bound on unmodeled dynamics, as established by the universal approximation.*


**Proposition** **2.**
*Assuming the presence of a positively definite function V(z) established within a vicinity surrounding zero for the nonlinear system (Equation 5), if V(0)=0 and its time-derivative adheres to the subsequent inequality:*

(35)
$$\dot{V}+\iota V^{l}-\varrho \leq 0$$

*where 0<l<1, ι, and ϱ are fixed positive values, then the system exhibits SGPFS behavior.*


**Theorem** **1.**
*Under Assumptions 1–3 and Proposition 1, the funnel backstepping control strategy, as presented in equation (Equation 18) with its robustifying term specified in (Equation 19) and estimator given in (Equation 23), coupled with the adaptation laws (Equation 20) and (Equation 21), guarantees the SGPFS of the system (Equation 5), while also ensuring that the tracking error (Equation 8) remains within the safe set.*


**Proof** **of** **Theorem** **1.**To derive a control law that can guarantee the stability of the first subsystem, the following Lyapunov function will be introduced as a potential candidate:
(36)$$V_{1}=\frac{1}{2}\sum_{i=1}^{i=n}x_{1i}^{2}$$
One way to stabilize the first subsystem is to propose the virtual input as ${\dot{x}}_{1i}=-k_{1i}x_{1i}^{2l-1}$. By implementing this virtual input, the time derivative of the Lyapunov function given in (Equation 36) will be in the following form:
(37)$${\dot{V}}_{1}=-\sum_{i=1}^{i=n}k_{1i}x_{1i}^{2l}\leq 0$$
Equation (Equation 37) implies that the first subsystem is semi-definite stable. In terms of the main subsystem, the strategy will be to first define a new variable as
(38)$$z_{i}=x_{2i}+k_{1i}x_{1i}^{2l-1}$$
and then, based on the newly formed state space representation, formulate the Lyapunov candidate for the second subsystem as:
(39)$$V_{2}=\sum_{i=1}^{i=n}\left[V_{1i}+0.5{\left(x_{2i}+k_{1i}x_{1i}^{2l-1}\right)}^{2}+\left(0.5/\alpha_{i}\right){\tilde{\rho }}_{i}^{2}+\left(0.5/\beta_{i}\right){\tilde{\theta }}_{i}^{⊤}{\tilde{\theta }}_{i}\right]$$
in which ${\tilde{\rho }}_{i}$ and ${\tilde{\theta }}_{i}$ are defined as ${\tilde{\rho }}_{i}=\rho_{i}-{\hat{\rho }}_{i}$ and ${\tilde{\theta }}_{i}=\theta_{i}-{\hat{\theta }}_{i}$, respectively. Here, $\rho_{i}$ denotes the actual parameter representing the upper bound of the unmodeled dynamics, while ${\hat{\rho }}_{i}$ represents the estimated value of this upper bound. Similarly, $\theta_{i}$ and ${\hat{\theta }}_{i}$ refer to the actual parameters and their estimations related to the modeled uncertain dynamics.To analyze the rate of energy changes within the Lyapunov function, one can take the time derivative of Equation (Equation 39) to obtain:
(40)$${\dot{V}}_{2i}=\sum_{i=1}^{i=n}\left[x_{1i}x_{2i}+\left(x_{2i}+k_{1i}x_{1i}^{2l-1}\right)\left({\dot{x}}_{2i}+K_{1i}\left(2l-1\right)x_{2i}x_{1i}^{2l-2}\right)-\left(1/\alpha_{i}\right){\tilde{\rho }}_{i}{\dot{\hat{\rho }}}_{i}-{\left(1/\beta_{i}\right){\tilde{\theta }}_{i}}^{⊤}{\dot{\hat{\theta }}}_{i}\right]$$
Subsequently, the dynamics of the robot, as described in (16), must be integrated into (Equation 40) to produce:
(41)$$\begin{array}{c} {\dot{V}}_{2i}=\sum_{i=1}^{i=n}\left[x_{1i}x_{2i}+\left(x_{2i}+k_{1i}x_{1i}^{2l-1}\right)\left({\hat{D}}_{i}^{-1}\tau_{i}+{\bar{\mu }}_{i}+K_{1i}\left(2l-1\right)x_{2i}x_{1i}^{2l-2}\right)\right]-\\ -\sum_{i=1}^{i=n}\left[{\left(1/\alpha_{i}\right)\tilde{\rho }}_{i}{\dot{\hat{\rho }}}_{i}+{\left(1/\beta_{i}\right){\tilde{\theta }}_{i}}^{⊤}{\dot{\hat{\theta }}}_{i}\right] \end{array}$$
Here, ${\bar{\mu }}_{i}$ represents the lumped uncertainty and can be modeled by the following equation:
(42)$${\bar{\mu }}_{i}={\theta_{i}}^{⊤}\gamma_{i}+\epsilon_{i}$$
In Equation (Equation 42), $\theta_{i}$, $\gamma_{i}$, and $\epsilon_{i}$ represent the uncertainty parameters, regressors, and unmodeled dynamics, respectively. To address this uncertainty, the approach involves utilizing a fuzzy system for estimation. In essence, the fuzzy system will be designed with an appropriate structure and inputs so that its output provides a suitable estimate of the uncertainty.By applying the control law $\tau_{i}$ defined in Equation (Equation 18) along with the robustifying term $\tau_{ri}$ as formulated in Equation (Equation 19), and through subsequent manipulation, the following components will be incorporated:
(43)$$\begin{array}{c} {\dot{V}}_{2i}=\sum_{i=1}^{i=n}\left[-k_{1i}x_{1i}^{2l}-k_{2i}{\left(x_{2i}+k_{1i}x_{1i}^{2l-1}\right)}^{2l}-{\left(x_{2i}+k_{1i}x_{1i}^{2l-1}\right)}^{2}{\hat{\rho }}^{2}{\left(|x_{2i}+k_{1i}x_{1i}^{2l-1}|\hat{\rho }+\delta_{ri}e^{-\sigma_{ri}t}\right)}^{-1}\right]\\ \sum_{i=1}^{i=n}\left[\left(x_{2i}+k_{1i}x_{1i}^{2l-1}\right)\left({{\tilde{\theta }}_{i}}^{⊤}\gamma_{i}+\epsilon_{i}\right)-{\left(1/\alpha_{i}\right)\tilde{\rho }}_{i}{\dot{\hat{\rho }}}_{i}-{\left(1/\beta_{i}\right){\tilde{\theta }}_{i}}^{⊤}{\dot{\hat{\theta }}}_{i}\right] \end{array}$$**Lemma** **1.**
*For any non-negative variables a, b, and δ, the following inequality holds:*

(44)
$$a^{2}b^{2}{\left(ab+\delta \right)}^{-1}\geq ab-\delta$$

By using the inequality given in Lemma 1, the following can be concluded:
(45)$$-{\left(x_{2i}+k_{1i}x_{1i}^{2l-1}\right)}^{2}{\hat{\rho }}^{2}{\left(|x_{2i}+k_{1i}x_{1i}^{2l-1}|\hat{\rho }+\delta_{ri}e^{-\sigma_{ri}t}\right)}^{-1}\leq -\left|x_{2i}+k_{1i}x_{1i}^{2l-1}\right|\hat{\rho }+\delta_{ri}e^{-\sigma_{ri}t}$$
Proposition 1 establishes that the unmodeled dynamics $\epsilon_{i}$ can be bounded provided that the fuzzy system is appropriately structured. This conclusion, along with considering $\rho_{i}$ as an upper bound for $\epsilon_{i}$, facilitates the derivation of the following inequality for all $x_{1}$ and $x_{2}$:
(46)$$\left(x_{2i}+k_{1i}x_{1i}^{2l-1}\right)\epsilon_{i}\leq \left|x_{2i}+k_{1i}x_{1i}^{2l-1}\right|\rho_{i}$$
By considering the worst-case scenario modeled with the upper bound defined in Equations (Equation 45) and (Equation 46), the exchange energy rate in the Lyapunov function, as defined in Equation (Equation 43), can be rewritten as follows:
(47)$$\begin{array}{c} {\dot{V}}_{2i}\leq \sum_{i=1}^{i=n}\left[-k_{1i}x_{1i}^{2l}-k_{2i}{\left(x_{2i}+k_{1i}x_{1i}^{2l-1}\right)}^{2l}+\delta_{ri}e^{-\sigma_{ri}t}+\left(x_{2i}+k_{1i}x_{1i}^{2l-1}\right){{\tilde{\theta }}_{i}}^{⊤}\gamma_{i}+\left|x_{2i}+k_{1i}x_{1i}^{2l-1}\right|\tilde{\rho_{i}}\right]\\ -\sum_{i=1}^{i=n}\left[{\left(1/\alpha_{i}\right)\tilde{\rho }}_{i}{\dot{\hat{\rho }}}_{i}+{\left(1/\beta_{i}\right){\tilde{\theta }}_{i}}^{⊤}{\dot{\hat{\theta }}}_{i}\right] \end{array}$$
By substituting (Equation 20) and (Equation 21) into (Equation 47) and performing some manipulation and simplification, the following can be achieved:
(48)$${\dot{V}}_{2i}\leq \sum_{i=1}^{i=n}\left[-k_{1i}x_{1i}^{2l}-k_{2i}{\left(x_{2i}+k_{1i}x_{1i}^{2l-1}\right)}^{2l}+\delta_{ri}e^{-\sigma_{ri}t}+{\left(\sigma_{\theta i}/\beta_{i}\right){\tilde{\theta }}_{i}}^{⊤}{\hat{\theta }}_{i}+\left(\sigma_{\rho i}/\alpha_{i}\right){\tilde{\rho }}_{i}{\hat{\rho }}_{i}\right]$$**Lemma** **2**([19])**.**
*For positive constants a, b, and c, and for any real variables $z_{1}$ and $z_{2}$ the subsequent inequality holds:*
(49)$${\left|z_{1}\right|}^{a}{\left|z_{2}\right|}^{b}\leq \frac{a}{a+b}c{\left|z_{1}\right|}^{a+b}+\frac{b}{a+b}c^{-a/b}{\left|z_{2}\right|}^{a+b}$$**Lemma** **3**([20])**.**
*Supposing $c_{1},c_{2},\cdots ,c_{n}$ are all positive numbers and 0<l<1, the following inequality remains valid:*
(50)$${\left(c_{1}+c_{2}+\cdots +c_{n}\right)}^{l}\leq c_{1}^{l}+c_{2}^{l}+\cdots +c_{n}^{l}$$In accordance with the worst-case scenario for the exchange energy change of the Lyapunov function, we consider the following equality:
(51)$$\sum_{j=1}^{j=M}{\tilde{\theta }}_{ij}{\hat{\theta }}_{ij}=\sum_{j=1}^{j=M}{\tilde{\theta }}_{ij}\theta_{ij}-\sum_{j=1}^{j=M}{\tilde{\theta }}_{ij}^{2}$$
On the other hand, the following inequality holds:
(52)$$-0.5\left({\tilde{\theta }}_{ij}^{2}+{\theta_{ij}}^{2}\right)\leq {\tilde{\theta }}_{ij}\theta_{ij}\leq 0.5\left({\tilde{\theta }}_{ij}^{2}+{\theta_{ij}}^{2}\right)$$
By using (Equation 52), the left hand side of Equation (Equation 51) can be written as:
(53)$$\sum_{j=1}^{j=M}{\tilde{\theta }}_{ij}{\hat{\theta }}_{ij}\leq 0.5\sum_{j=1}^{j=M}\theta_{ij}^{2}-0.5\sum_{j=1}^{j=M}{\tilde{\theta }}_{ij}^{2}$$
Similarly, the following inequality can be concluded:
(54)$${\tilde{\rho }}_{i}{\hat{\rho }}_{i}\leq 0.5\rho_{i}^{2}-0.5{\tilde{\rho }}_{i}^{2}$$
Upon substituting (Equation 53) and (Equation 54) into (Equation 48), the following is obtained:
(55)$$\begin{array}{c} {\dot{V}}_{2i}\leq \sum_{i=1}^{i=n}\left[-k_{1i}x_{1i}^{2l}-k_{2i}{\left(x_{2i}+k_{1i}x_{1i}^{2l-1}\right)}^{2l}-{\left(\sigma_{\theta i}/2\beta_{i}\right){\tilde{\theta }}_{i}}^{⊤}{\tilde{\theta }}_{i}-{\left(\sigma_{\rho i}/2\alpha_{i}\right){\tilde{\rho }}_{i}}^{2}\right]\\ +\sum_{i=1}^{i=n}\left[{\left(\sigma_{\theta i}/2\beta_{i}\right)\theta_{i}}^{⊤}\theta_{i}+{\left(\sigma_{\rho i}/2\alpha_{i}\right)\rho_{i}}^{2}+\delta_{ri}e^{-\sigma_{ri}t}\right] \end{array}$$
By using Lemma 2, and defining 0<l<1, a=1−l, b=l, $z_{1}=1$, $z_{2}={\tilde{\theta }}_{ij}^{2}$, $c_{1}=l^{\frac{l}{\left(1-l\right)}}$, one can conclude that:
(56)$${\left({\tilde{\theta }}_{ij}^{2}\right)}^{l}\leq \left(1-l\right)c_{1}+{\tilde{\theta }}_{ij}^{2}$$
Then the following can be concluded:
(57)$$-{{\tilde{\theta }}_{i}}^{⊤}{\tilde{\theta }}_{i}\leq -{\left({{\tilde{\theta }}_{i}}^{⊤}{\tilde{\theta }}_{i}\right)}^{l}+\left(1-l\right)c$$
Similarly, one can derive the following inequality:
(58)$$-{\tilde{\rho }}_{i}^{2}\leq -{\left({\tilde{\rho }}_{i}^{2}\right)}^{l}+\left(1-l\right)c$$
Substituting (Equation 57) and (Equation 58) into (Equation 55) yields:
(59)$$\begin{array}{c} {\dot{V}}_{2i}\leq \sum_{i=1}^{i=n}\left[-k_{1i}x_{1i}^{2l}-k_{2i}{\left(x_{2i}+k_{1i}x_{1i}^{2l-1}\right)}^{2l}-{\left(\sigma_{\theta i}/2\beta_{i}\right)({\tilde{\theta }}_{i}}^{⊤}{\tilde{\theta }}_{i}{{)}^{l}-{\left(\sigma_{\rho i}/2\alpha_{i}\right)({\tilde{\rho }}_{i}}^{2})}^{l}\right]\\ +\sum_{i=1}^{i=n}\left[{\left(\sigma_{\theta i}/2\beta_{i}\right)\theta_{i}}^{⊤}\theta_{i}+{\left(\sigma_{\rho i}/2\alpha_{i}\right)\rho_{i}}^{2}+\delta_{ri}e^{-\sigma_{ri}t}+\left(1-l\right)c\left(\sigma_{\theta i}/2\beta_{i}+\sigma_{\rho i}/2\alpha_{i}\right)\right] \end{array}$$
Using Lemma 3 in (Equation 59) implies that
(60)$$\begin{array}{c} {\dot{V}}_{2i}\leq \sum_{i=1}^{i=n}\left[-{\left(k_{1i}x_{1i}^{2}+k_{2i}{\left(x_{2i}+k_{1i}x_{1i}^{2l-1}\right)}^{2}+{\left(\sigma_{\theta i}/2\beta_{i}\right)({\tilde{\theta }}_{i}}^{⊤}{\tilde{\theta }}_{i})+{\left(\sigma_{\rho i}/2\alpha_{i}\right)({\tilde{\rho }}_{i}}^{2})\right)}^{l}\right]\\ +\sum_{i=1}^{i=n}\left[{\left(\sigma_{\theta i}/2\beta_{i}\right)\theta_{i}}^{⊤}\theta_{i}+{\left(\sigma_{\rho i}/2\alpha_{i}\right)\rho_{i}}^{2}+\delta_{ri}e^{-\sigma_{ri}t}+\left(1-l\right)c\left(\sigma_{\theta i}/2\beta_{i}+\sigma_{\rho i}/2\alpha_{i}\right)\right] \end{array}$$
To simplify the equation further, let us designate ι as the minimum among the four coefficients below, following the strategy of designing for the worst-case scenario.
(61)$$\iota =\min\left\{2k_{1i},2k_{2i},\sigma_{\theta i},\sigma_{\rho i}\right\}$$
Regarding the remaining components within (Equation 61), let us consider them collectively as follows:
(62)$$\varrho =\sum_{i=1}^{i=n}\left[{\left(\sigma_{\theta i}/2\beta_{i}\right)\theta_{i}}^{⊤}\theta_{i}+{\left(\sigma_{\rho i}/2\alpha_{i}\right)\rho_{i}}^{2}+\delta_{ri}e^{-\sigma_{ri}t}+\left(1-l\right)c\left(\sigma_{\theta i}/2\beta_{i}+\sigma_{\rho i}/2\alpha_{i}\right)\right]$$
By employing Equations (Equation 61) and (Equation 62), the final expression for the Lyapunov exchange energy can be represented by the following mathematical equation:
(63)$${\dot{V}}_{2i}+\iota^{l}V_{2i}^{l}-\varrho \leq 0$$
By comparing this to Proposition 2, the SGPFS of the main subsystem will be achieved. □

### 4.2. Internal Behavior of the System

In the preceding subsection, the establishment of SGPFS was discussed. This establishment, as delineated by (Equation 9), ensures that if the initial condition falls within the task space funnel, the system’s energy undergoes decay and can be confined within a predefined funnel-shaped energy boundary. Consequently, Equation (Equation 63) asserts the boundedness of the components of the Lyapunov function, namely $x_{1i}$, $x_{2i}$, ${\tilde{\theta }}_{i}$, and ${\tilde{\rho }}_{i}$. This boundedness extends to the sliding surface $S_{i}$ and its time derivative ${\dot{S}}_{i}$. The consequent boundedness of $S_{i}$ and ${\dot{S}}_{i}$ implies the boundedness of $\xi_{i}$ and ${\dot{\xi }}_{i}$. By selecting positive values for $\beta_{i}$, $\alpha_{i}$, $\sigma_{\theta_{i}}$, and $\sigma_{\rho_{i}}$, it is inferred that ${\hat{\theta }}_{i}$ and ${\hat{\rho }}_{i}$ are also bounded in accordance with (Equation 20) and (Equation 21). Additionally, the fuzzy regressor $\gamma_{i}$, which is a function of $x_{1i}$ and $x_{2i}$, is bounded when considering the normalized membership function condition alongside the completeness of the fuzzy system. Consequently, all signals on the right-hand side of (Equation 23) exhibit bounded behavior, thereby implying the boundedness of ${\hat{\bar{\mu }}}_{i}$.

By designing the robot’s desired trajectory to circumvent singularity points (i.e., ensuring the existence of the inverse of the Jacobian matrix), it can be demonstrated that the robot variable $q_{i}$ and its time derivative also remain bounded.

Likewise, the boundedness of all components on the right-hand side of Equations (Equation 18) and (Equation 19) indicates the boundedness of the robust controller $\tau_{ri}$ and the main controller $\tau_{i}$. Utilizing Proposition 1 and substituting the bounded (Equation 18) and (Equation 19) into (Equation 5), it can be concluded that the acceleration signal ${\ddot{q}}_{i}$ is also bounded.

The boundedness of $q_{i}$ implies the boundedness of $p_{i}$, considering the upper bound of the rotation matrix. This boundedness of $p_{i}$ extends to the boundedness of *x*, *y*, and ψ of the robot. Similarly, it can be inferred that ${\dot{p}}_{i}$ is also bounded. Referring to (Equation 1), the boundedness of ${\dot{p}}_{i}$, along with the Assumptions 1 and 3, and considering the upper-boundedness of external disturbances in Kinematics, leads to the conclusion that the internal velocity signals $u_{r}$ and *r* are also bounded. Referring to (2), the boundedness of ${\dot{u}}_{r}$ and $\dot{r}$ can be deduced under the assumption of upper-bounded external disturbances in Dynamics and lateral velocity, as well as other upper-bounded signals in (2).

## 5. Simulation Results and Discussion

For the effectiveness assessment of the proposed method and comparison, MATLAB Version 23.2 (R2023b) was employed.

### 5.1. Simulation A: Proposed Method

Here, the developed scheme is simulated on the robot in the presence of important uncertainties. The dynamics of an under-actuated marine vehicle navigating in two dimensions were analyzed using the information given in Table 1 [21].

In the MATLAB setup, the sampling time was configured as $T_{s}=0.001$ s, and the program was executed for $t_{f}=1800$ s. The intended trajectory for the robot within the two-dimensional plane is expressed through the following set of equations: (64)$$x_{d}\left(t\right)=\left\{\begin{array}{cc} 0.5t & \mathrm{if}t<Tc_{1},\\ \left(Ra-Ts/60\right)\cos(\pi \left(t-Tc_{1}\right)/Wc)\; & \mathrm{if}Tc_{1}\leq t<Tc_{2},\\ -80 & \mathrm{if}t\geq Tc_{2}. \end{array}\right.$$
(65)$$y_{d}\left(t\right)=\left\{\begin{array}{cc} 0 & \mathrm{if}t<Tc_{1},\\ \left(Ra-Ts/60\right)\sin(\pi \left(t-Tc_{1}\right)/Wc)\; & \mathrm{if}Tc_{1}\leq t<Tc_{2},\\ -0.4(t-Tc_{2}) & \mathrm{if}t\geq Tc_{2}. \end{array}\right.$$
where $Tc_{1}$, $Tc_{2}$ and Wc are defined as $Tc_{1}=200$, $Tc_{2}=1400$, Wc=400. The trajectory consists of three distinct sub-trajectories: the first being a linear segment for $t<Tc_{1}$, followed by a rotational trajectory with a time-varying radius for $Tc_{1}\leq t<Tc_{2}$, and concluding with another linear segment for $t\geq Tc_{2}$. The parameters of the proposed control are illustrated in Table 2.

The transition between linear and rotational trajectories is deliberately designed to occur abruptly, resulting in a discontinuity where the first-time derivative does not exist. This intentional design choice serves to simulate scenarios where significant challenges arise, allowing for the effective evaluation of the system’s performance and stability. An instance of such a scenario could occur in the presence of underwater turbulence or during obstacle avoidance maneuvers.

When confronted with the erratic and irregular movement of water or the need to avoid collisions with the environment, the system must respond promptly based on the data collected by safety sensors. One approach to mitigating the adverse effects of sudden changes in trajectory is to rapidly adjust the desired trajectory. However, this adjustment may sacrifice smoothness, potentially necessitating points of non-differentiability at transition points, denoted as $Tc_{1}$ and $Tc_{2}$. The primary controller must be adequately equipped to address this challenge and undergo thorough evaluation and testing to ensure its effectiveness.

The robot experiences external disturbances affecting its Kinematics, with the components represented by $\tau_{d_{kx}}=\tau_{d_{ky}}=\tau_{d_{kr}}=0.05\sin\;\left(0.1t-10\right)$. In terms of dynamic uncertainties, signals for $\tau_{d_{u_{r}}}$, $\tau_{d_{v_{r}}}$, and $\tau_{d_{r}}$ are expressed as $\tau_{d_{u_{r}}}=10\sin\;\left(0.1t-10\right)$ and $\tau_{d_{v_{r}}}=\tau_{d_{r}}=0.2\sin\;\left(0.1t-10\right)$. The desired and actual trajectories of the robot, along with a schematic representation of the robot indicating its orientation, are depicted in Figure 3. The robot can initially start from any arbitrary position and heading angle; the only condition is that the robot begins its movement inside the funnel. In this simulation, the marine robot initially starts with a heading angle of $\psi_{0}=\pi /6$, and its starting position is $p_{0}={\left[\begin{array}{cc} -1.2 & -8 \end{array}\right]}^{⊤}m$. The heading angle is separately depicted in Figure 4. The change in the heading angle and its oscillatory behavior enable the robot to generate the necessary force for the engine to execute the maneuvers required to track the trajectory, even as unwanted disturbances affect the robot, attempting to divert it from the desired path. The robot successfully tracks the desired trajectory while reducing its error within a finite time frame. The robot’s task space comprises four regions: $Z_{1}$, $Z_{2}$, $Z_{3}$, and $Z_{4}$, representing scenarios with varying water current speeds that may endanger system stability. The water current speed in each zone is determined by the following equations:$Z_{1}:v_{c1}={\left[\begin{array}{cc} -0.1\cos(0.1t) & +0.2\sin(0.1t) \end{array}\right]}^{⊤}$;$Z_{2}:v_{c2}={\left[\begin{array}{cc} +0.2\cos(0.1t) & -0.1\sin(0.1t) \end{array}\right]}^{⊤}$;$Z_{3}:v_{c3}={\left[\begin{array}{cc} +0.4\cos(0.1t) & +0.2\sin(0.1t) \end{array}\right]}^{⊤}$;$Z_{4}:v_{c4}={\left[\begin{array}{cc} +0.3\cos(0.1t) & -0.1\sin(0.1t) \end{array}\right]}^{⊤}$.

For better clarity, Figure 3 and Figure 4 are color-coded based on entry into each zone, making it easier to observe the behavior of the system. The tracking errors of the robot along the *x* and *y* directions, as well as the safe area, are illustrated in Figure 5.

Thanks to the robust funnel control strategy proposed in this paper, the tracking error consistently stays within the boundaries of the funnel shape. However, the robot must commence its movement within the funnel for this strategy to take effect.

Despite encountering abrupt changes in the desired path at times t=200 and t=1400 s, as well as sudden alterations in the amplitude and direction of the water current speeds at the boundaries, which attempted to push the error beyond the funnel boundaries, the controller successfully coerced the system back inside the safe set, as demonstrated in the Figure 5. Ensuring that the tracking error remains within the funnel boundary is crucial for guaranteeing the safety of the controller. By observing the performance of the robot as depicted in fig3,fig4,fig5, one can conclude that the controller effectively compensates for external disturbances affecting the robot, as well as various changes in water current speed that pose a threat to the robot’s stability.

The control signals $\tau ={\left[\begin{array}{cc} \tau_{u_{r}} & \tau_{r} \end{array}\right]}^{⊤}$ are plotted in Figure 6 and have the potential to be directly applied to the robot’s actuators. By exerting these two signals, namely $\tau_{u}$ and $\tau_{r}$, to the robot actuators ([Disp-formula FD2a-sensors-24-03974]) and (2c), respectively, the robot can track the desired trajectory in task space in the presence of uncertainties. Notably, Equation (2b) illustrates the equation of motion along the lateral velocity, where the absence of control input, as indicated by the right-hand side being zero, signifies the lack of control in this direction, that is, the vehicle is underactuated. In our proposed methodology, we leverage $\tau_{u}$ and $\tau_{r}$ as the only control signals that are given by the control law (Equation 18).

To gain a better understanding of the internal behavior of the system, Figure 7 shows the linear and angular velocity signals of the robot which help us in assessing the feasibility of implementation on a real robot in experimental tests. As it can be seen, the robot exhibits good performance, even in the presence of significant sources of uncertainty.

The adaptation of fuzzy parameter estimation ${\hat{\theta }}_{ij}$ is illustrated in Figure 8. Considering three membership functions for each of the two fuzzy inputs, $x_{1}$ and $x_{2}$, nine fuzzy rules, which correspond to the number of parameters, form the rule base of the estimator along each task space direction, *x* and *y*. Two fuzzy estimators are required: one for compensating uncertainties along the *x* direction and the other along the *y* direction. The uncertainty upper bound estimation, represented as ${\hat{\rho }}_{i}$, is depicted in Figure 9.

The rapid dynamical behavior of certain fuzzy parameters ${\hat{\theta }}_{i}$ and the robust term ${\hat{\rho }}_{i}$ enables the controller to react quickly and effectively to external disturbances at both the Kinematics and Dynamics levels.

### 5.2. Simulation B: Gradient-Descent Based Control

For comparison purposes, the methodology described in [15] has been implemented under identical simulation conditions. In order to maintain methodological continuity and emphasize the significant improvements achieved, we have chosen to primarily compare our proposed method with our previous one. This facilitates a clearer demonstration of the improvement and novelty of the proposed method as an extended version of the previous work. For further information regarding the controller and its structure, please refer to [15].

The tracking errors, control efforts exerted on robot actuators, and velocity control laws provided by the outer loop are illustrated in Figure 10, Figure 11 and Figure 12, respectively.

According to Figure 10, it is evident that the tracking error may not necessarily have remained within the safe area. Consequently, it cannot be deemed as a safe controller.

### 5.3. Comparative Analysis and Interpretation

In this subsection, we delve deeper into the simulation results obtained with the proposed method discussed in Section 5.1, and the comparative method analyzed in Section 5.2 as our previous work presented in [15]. This comparison serves two primary objectives. Firstly, it aims to underscore the critical importance of designing a controller that ensures safety. Secondly, it endeavors to achieve a single-loop, direct control over actuators through the application of advanced nonlinear control techniques. Such techniques are poised to elicit more robust behaviors in the face of various potential sources of uncertainty that could jeopardize system stability.

To facilitate comparison, Figure 13 presents the tracking error norms for both the proposed method and the gradient descent-based controller.

Upon analyzing this signal, it becomes evident that the proposed method significantly outperforms the gradient descent fuzzy estimator method.

Based on the tracking error evolution signals depicted in Figure 5 and Figure 10, one can observe that the proposed method, designed with safety measures in mind, outperforms [15]. To be more specific, the tracking error in Figure 10 exceeded the safe boundary $-\nu_{i}$ at t=88 for the x-direction and at t=61 seconds for the y-direction. It remained outside this threshold for a significant duration. Conversely, the tracking error evolution of the proposed method demonstrated consistency in remaining within safe bounds.

In terms of convergence speed, a significant difference can be seen between the two approaches, as depicted in Figure 13. This difference can be attributed to the two-loop control in [15] versus the direct single-loop control on actuators in the proposed method. It is imperative to clarify that the signals $u_{r}$ and *r* showcased in Figure 7 are not intended as control signals. In contrast, the gradient descent-based method, serving as our comparative benchmark, employs all four signals—$\tau_{u}$, $\tau_{r}$, $u_{r}$, and *r*—in two distinct control loops. This dual-loop configuration necessitates that the control signals $u_{r}$ and *r*, generated by the outer loop, be fed into the inner loop as commands.

A comparative analysis of the results reveals distinct patterns in the performance of our proposed method compared to the comparative approach. Key differences in trajectory tracking accuracy, convergence speed, and robustness to disturbances are observed between the two methods. These results underscore the superiority of our proposed method, highlighting the efficacy of a single-loop approach with direct control over actuators, as opposed to our prior methodology. Furthermore, it can provide assurance of their stability and safety evolution, and by utilizing feasible sensory feedback and adopting a pessimistic approach towards surrounding uncertainties, their viability for real-world implementation can be substantiated.

## 6. Conclusions

In this paper, we introduced a novel funnel backstepping controller designed to ensure the safe navigation of an underactuated marine robot, with formal stability guarantees incorporated into its design. We have tackled major challenges from ocean conditions directly affecting the robot’s stability. The controller’s structure, coupled with its fuzzy estimator, is simplistic and integrates straightforward sensory data feedback mechanisms. These attributes render it highly suitable for practical implementation in real-world settings. Our simulation results and comparative analyses demonstrate the effectiveness, robustness, and safety enhancements achieved by the robot, as evidenced by the visualization of internal signal plots. These simulations encompassed a spectrum of challenging scenarios, including time-varying external disturbances, unmodelled dynamics, fluctuating water currents, and lateral velocities. The proposed safe backstepping control mechanism demonstrated robustness in navigating intricate marine conditions, supported by comprehensive comparative analysis.

## Figures and Tables

**Figure 1 sensors-24-03974-f001:**
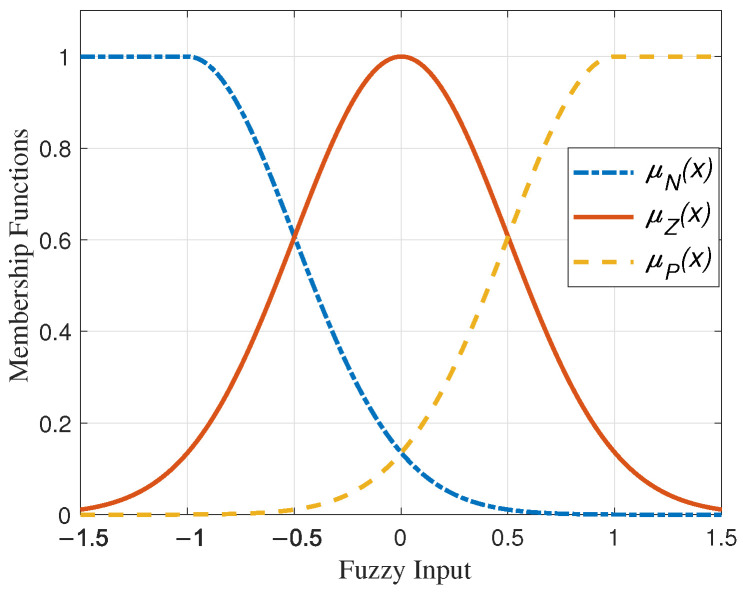
Membership functions of fuzzy systems.

**Figure 2 sensors-24-03974-f002:**
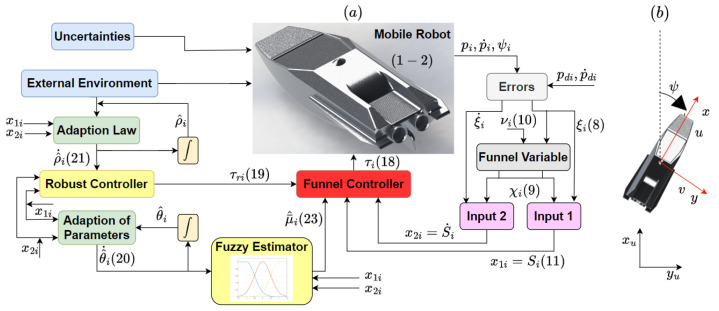
(**a**) Block diagram illustrating the developed scheme. (**b**) Robot coordinate frames in task space.

**Figure 3 sensors-24-03974-f003:**
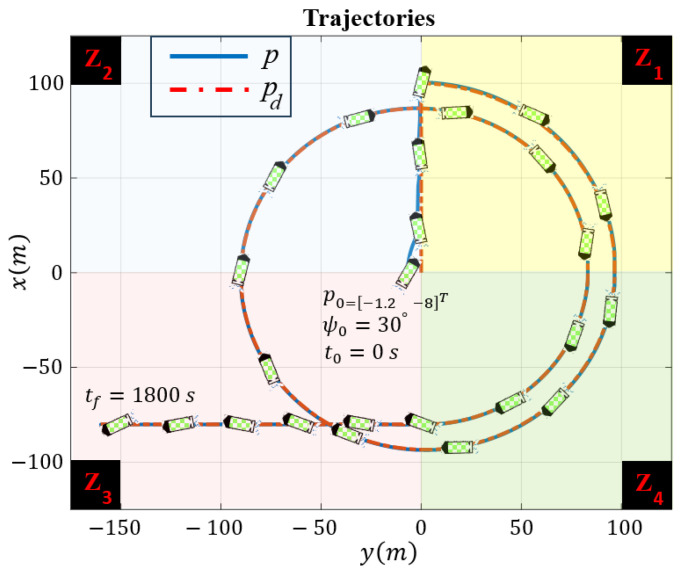
Trajectories of the mobile robot , with each zone color-coded to indicate entry into different areas.

**Figure 4 sensors-24-03974-f004:**
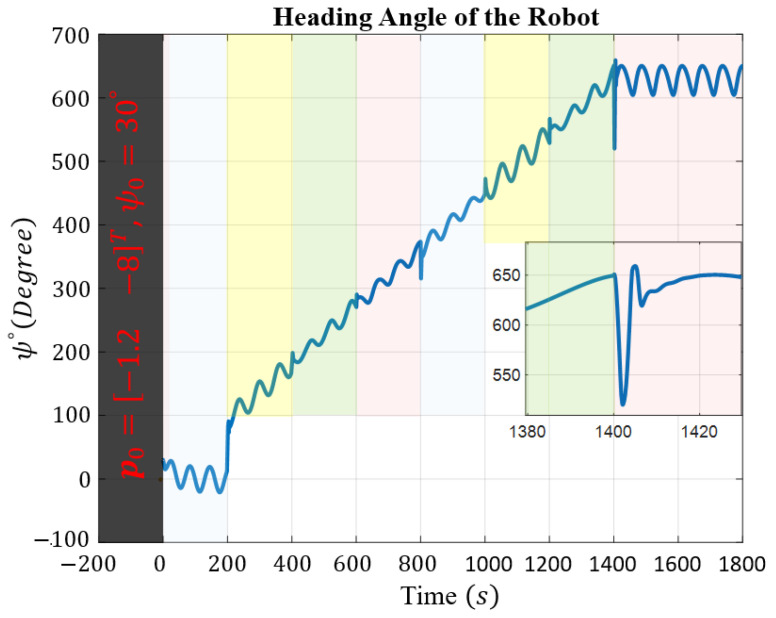
Heading angle of the mobile robot, with each zone color-coded to indicate entry into different areas.

**Figure 5 sensors-24-03974-f005:**
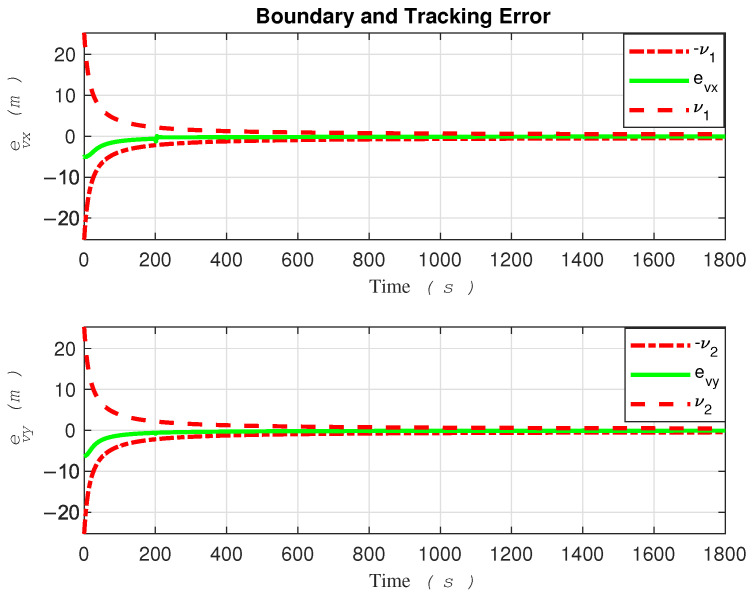
Error evolution in the proposed funnel control.

**Figure 6 sensors-24-03974-f006:**
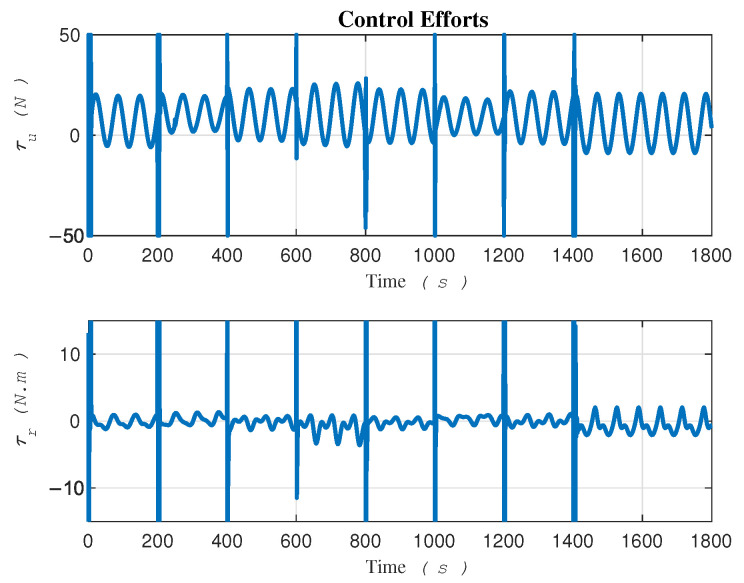
Control signals exerting to robot’s actuators.

**Figure 7 sensors-24-03974-f007:**
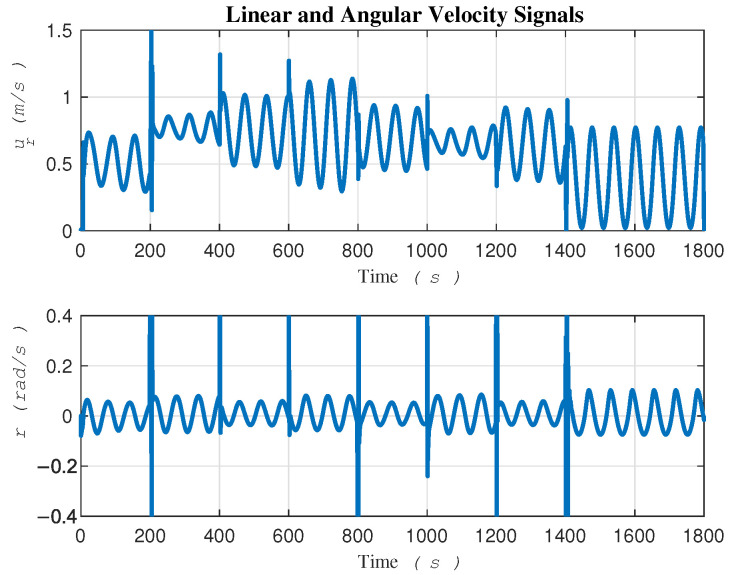
Internal behavior of the system.

**Figure 8 sensors-24-03974-f008:**
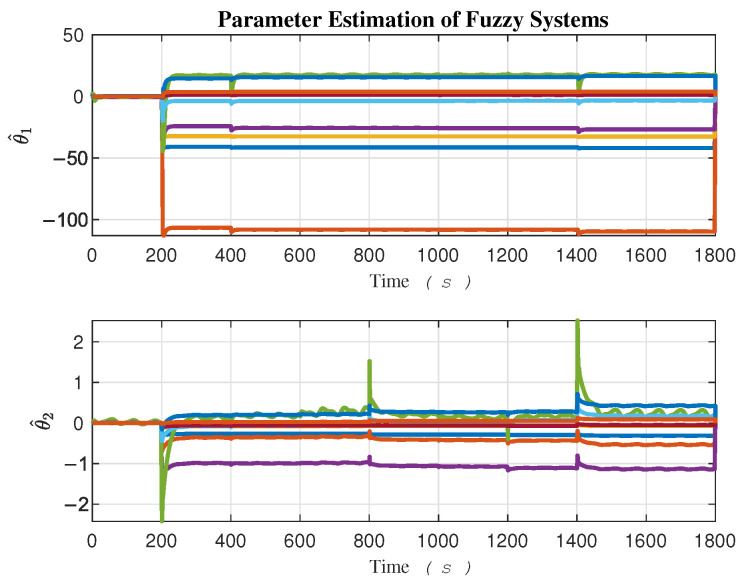
Adaption parameters of the fuzzy estimators.

**Figure 9 sensors-24-03974-f009:**
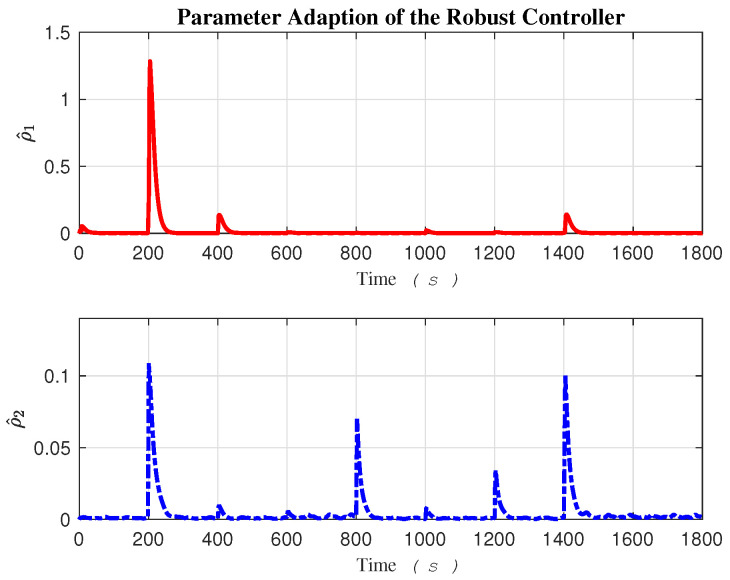
Adaption parameter of the robust control.

**Figure 10 sensors-24-03974-f010:**
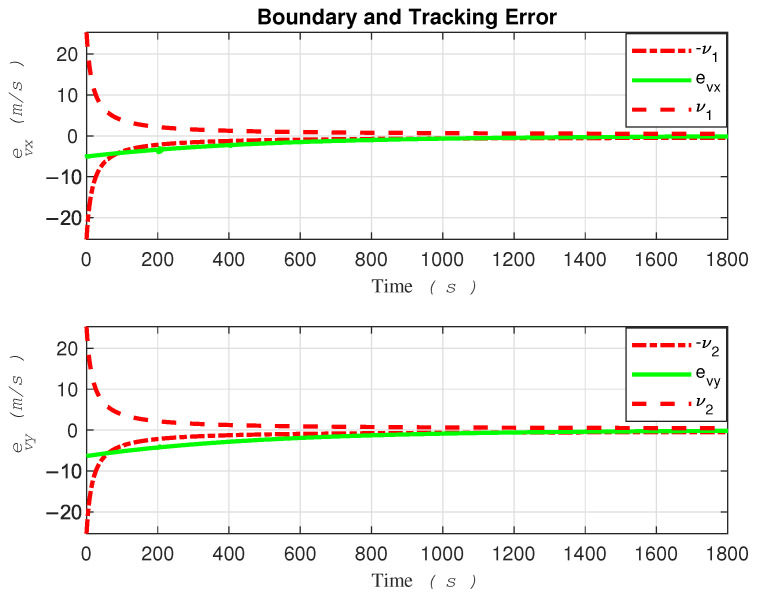
Error evolution in the gradient-descent control [15].

**Figure 11 sensors-24-03974-f011:**
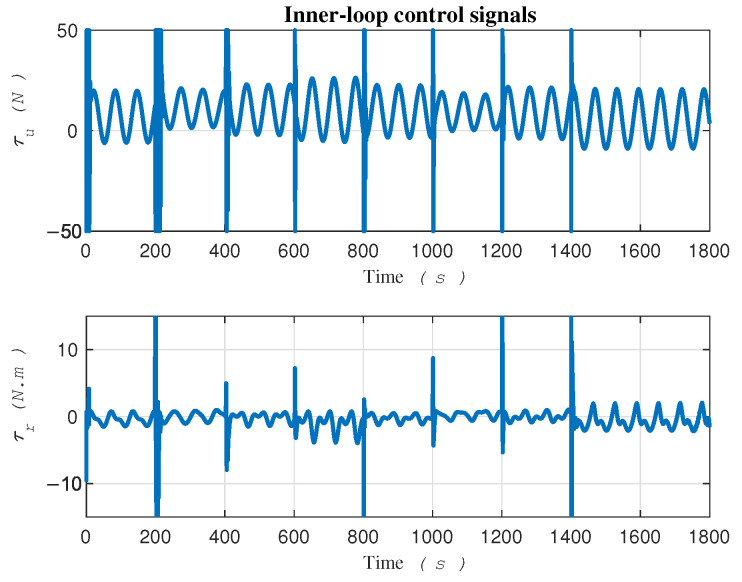
Inner-loop control signals in the gradient-descent approach [15].

**Figure 12 sensors-24-03974-f012:**
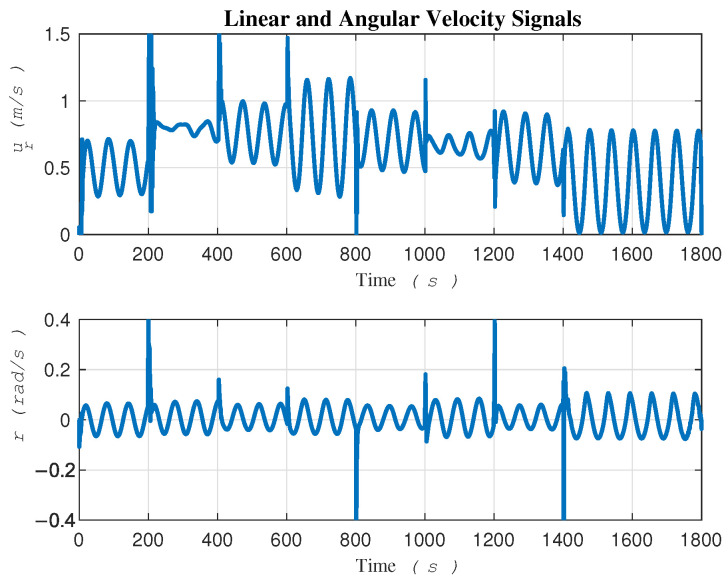
Outer-loop control signals in the gradient-descent approach [15].

**Figure 13 sensors-24-03974-f013:**
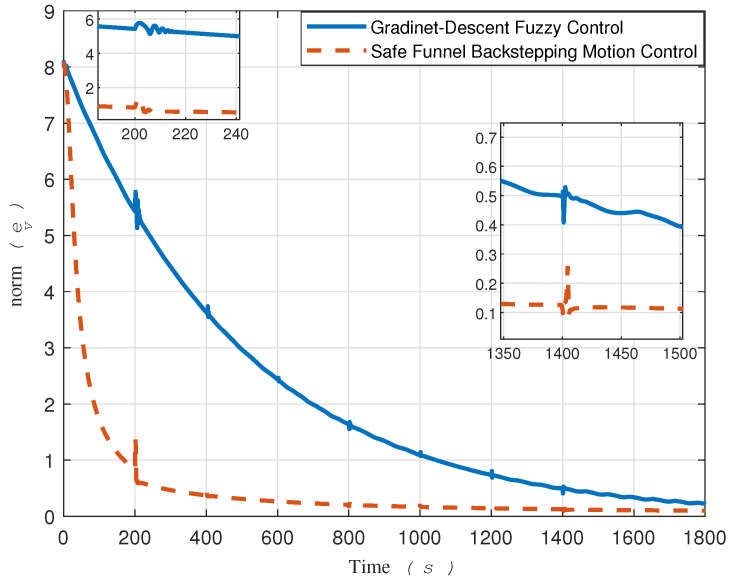
Comparison of the two norm errors of the proposed and gradient-descent-based control.

**Table 1 sensors-24-03974-t001:** Dynamic parameters of the mobile robot.

$m_{u}\;\left(\mathrm{kg}\right)$	$m_{v}\;\left(\mathrm{kg}\right)$	$m_{r}\;\left(\mathrm{kg}\cdot m^{2}\right)$	$d_{\mathrm{ur}}\;\left(\mathrm{kg}/s\right)$	$d_{\mathrm{vr}}\;\left(\mathrm{kg}/s\right)$	$d_{r}\;\left(\mathrm{kg}\cdot m^{2}/s\right)$
47.52	104.05	13.38	13.5	44.96	27.2

**Table 2 sensors-24-03974-t002:** Proposed control method specifications.

$\varrho_{0}$	$\varrho_{\infty }$	$\zeta_{i}$	β	$k_{1i}$	$k_{2i}$	$\delta_{\mathrm{ri}}$	$\sigma_{\mathrm{ri}}$	$\beta_{1}$	$\beta_{2}$	$\sigma_{\theta i}$	$\alpha_{i}$	$\sigma_{\rho i}$
25	0.25	0.8	0.06	8	39.5	0.1	0.2	20	8	0.0005	0.1	0.1

## Data Availability

The data presented in this study are available on request from the corresponding author.

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
