# Peer review of "Safe Robust Adaptive Motion Control for Underactuated Marine Robots"

_sensors, 2024, doi:10.3390/s24123974_

Round 1
Reviewer 1 Report
Comments and Suggestions for Authors
This is an excellent research paper. The overall quality of the paper is very high, but there are several small problems as follows:
1.Adjust the size of the chart properly to avoid excessive blank pages of the paper, for example, there is a blank page on page 21 that is not filled with text or charts.
2. In Figure 6 on page 17, why choose this type of control signal? A corresponding explanation or explanation is required.
3. For the simulation part, it is suggested to analyze the simulation results more carefully and carry out more thorough discussion and analysis. If possible, it is recommended to add more comparisons between different methods to highlight the high performance of the methods presented in this paper.
Reviewer 2 Report
Comments and Suggestions for Authors
This is an interesting study and the authors have proposed a safe adaptive backstepping control system for underactuated marine robots, utilizing straightforward sensors to enhance design simplicity and feasibility. The control system ensures robustness against uncertainties through a straightforward fuzzy subsystem and a funnel-based nonlinear sliding surface. Analysis of the exchange energy rate in the Lyapunov function demonstrated the Semi-Globally Practically Finite-Time Stability of the system, and simulation results further validated its safe controllability enabling quick and effective reaction to external disturbances.
In sum, this paper has novel viewpoints and rigorous demonstrations, and is generally well-written and structured. I think it is suitable for Sensors and of interest to the community. However, before submitting the final files, I recommend a few minor revisions based on the following suggestions, which I believe can help attract a broader readership.
1. In Section 2, the authors modeled the kinematics and dynamics of the marine robot, and the displacements and the heading angle were presented. However, only the displacement errors were used in the proposed controller. I think it’s better to explain why the heading angle error was not used.
2. I recommend a revision of Figure 2 as there are many blank areas.
3. In Section 5, why was this initial position and heading angle chosen in the simulation? Are there any reasons or restrictions? It’s better to give an explanation.
4. In Section 5.2, the authors compared the proposed control method with their previous one and proved its superiority. Why not compare it with more other methods? Is it because comparisons have been made in the authors’ previous work? I think it’s better to give an explanation.
5. Some typos need to be corrected:
l Line 296: The defined symbol “ω” does not agree with the symbol “ι” in the equation (61)
l Figure 7 &12: The formatting of the units on the vertical axis of the charts is confused
